# Beta-Blocker Separation on Phosphodiester Stationary Phases—The Application of Intelligent Peak Deconvolution Analysis

**DOI:** 10.3390/molecules28073249

**Published:** 2023-04-05

**Authors:** Oktawia Kalisz, Mikołaj Dembek, Sylwia Studzińska, Szymon Bocian

**Affiliations:** Chair of Environmental Chemistry and Bioanalytics, Faculty of Chemistry, Nicolaus Copernicus University, 7 Gagarin St., 87-100 Toruń, Poland

**Keywords:** liquid chromatography, beta-blocker, separation, stationary phases, ChromSword, peak deconvolution analysis, i-PDeA II

## Abstract

Beta-blockers are a class of medications predominantly used to manage abnormal heart rhythms. They are also widely used to treat high blood pressure. From the liquid chromatography separation point of view, beta-blockers are interesting molecules due to their hydrophobic–hydrophilic properties. Thus, the study aimed to investigate the beta-blocker separation selectivity on four phosphodiester stationary phases in reversed-phase liquid chromatography (RP LC) and hydrophilic interactions liquid chromatography (HILIC). On tested stationary phases, beta-blockers provide retention in both chromatographic systems, RP LC and HILIC. Additionally, it was found that cation-exchange mechanisms have a significant contribution to retention. Separations were enhanced by applying ChromSword software for gradient optimization and Intelligent Peak Deconvolution Analysis to separate unseparated peaks digitally.

## 1. Introduction

Beta-adrenergic blockers (β-blockers) represent an important class of drugs used to treat cardiac diseases, which are a problem in approximately one-third of the worldwide population [1,2,3]. At least 20 β-blockers are now commonly used, e.g., metoprolol, atenolol, propranolol, alprenolol, carvedilol, etc. [4,5,6,7,8,9,10]. On the other hand, long-term treatment with β-blockers might induce depression and consequently the risk of suicide [8]. Reliable methods of their analysis are indispensable, especially for determining their purity, for pharmacokinetic and pharmacodynamic studies, metabolism studies (several metabolites are pharmacologically active and harmful), or even for doping control, since some β-blockers are prohibited in athletic competitions [1,4,6,7,9]. Moreover, β-blockers are used to reduce morbidity in animals during their transportation; consequently, these drugs are present in meat or milk [11]. Appropriate methods of analysis of these compounds are needed for standard substances, drug-active substances, and biological samples (blood, plasma, urine) [1,4,6,7,9,12,13]. Their analysis may provide useful information for clinical studies. For these reasons, β-blocker drug testing requires methods of high efficiency and selectivity in a short time.

Reversed-phase high-performance liquid chromatography with UV, fluorescence, or mass spectrometry detection has become the so-called ‘gold standard’ technique for the separation, qualification, and quantification of various β-blockers. However, some hydrophilic interaction liquid chromatography methods also showed excellent separation efficiency and selectivity [9,14]. In most cases, separation has been performed on alkyl stationary phases, such as C18 and C8. The cyano stationary phases and unmodified silica-based columns have also been used [4,5,6,9,11,12], as well as monolithic ones [7]. Mobile phases applied to β-blocker analysis usually consist of different combinations of acetonitrile or methanol with buffers. Phosphate buffers have been used most frequently (5 mM to 50 mM) [5,6,7,10]. However, sodium chloride, sodium perchloride, ammonium formate, and ammonium acetate are used. The ion-pair reagents (sodium dodecyl sulfate with tetrabutylammonium dihydrogen phosphate) were also utilized [4,6,7,9,12]. Additionally, mixtures of water and acetonitrile with the addition of acetic or formic acid have been applied, providing satisfactory β-blocker resolution when MS detection was applied [9,11].

As usual, the optimization of RP HPLC analysis is the key to obtaining complete separation of β-blocker mixtures. The literature shows that these compounds’ most influential chromatographic parameters are the mobile phase composition (especially in buffer selection), pH, flow rate, and temperature [1,7,8,9,12]. β-blockers are protonated at low pH of the mobile phase [7]. Usually, the increase in β-blocker retention is observed when the pH of the mobile phase is increased to 6.5. The pH increase reduces the protonation, increasing the hydrophobicity [7,9]. Another critical parameter is the type of stationary phase and its particle size. Four compounds were separated in 25 min using 5 µm C18; however, changing the column to monolithic allowed the time to be reduced to 5 min (with an increased flow rate) [4,7]. Reducing the particle size of the C18 stationary phase to 3.5 µm allows the separation of a 5–8 component mixture of β-blockers in 10 min [6], while a further reduction to 1.7 µm provides separation of up to a dozen compounds in the same time [12].

Stationary phases with incorporated polar groups mixed with the hydrophobic alkyl ligands, so-called polar-embedded stationary phases, are promising chromatographic materials [15,16]. Such materials containing both hydrophobic and hydrophilic ligands can be applied in reversed-phase liquid chromatography (RP LC). They can also be used in hydrophilic interaction liquid chromatography (HILIC). Depending on the mobile phase pH, mixed-mode stationary phases may be ionized. [17,18]. They may separate both polar and nonpolar analytes [19]. It was recently proven that polar-embedded stationary phases allow chromatographic elution and separation using pure water as a mobile phase [20]. These stationary phases are called mixed-mode stationary phases.

In many cases, the complete separation of the mixture is difficult, especially when separated compounds have similar structures. It is a common problem in the pharmaceutical industry. For this reason, many solutions are being developed to facilitate method optimization and data analysis. One is peak deconvolution analysis with photodiode array (PDA) detectors that allow using 3D PDA data [21]. Using a unique software function, it can separate peaks that are not resolved on-column. It offers better detection results and minimizes method development and analysis time [22].

The study aimed to characterize the selectivity of four phosphodiester stationary phases for separating beta-blockers. The research was enhanced by applying Peak Deconvolution Analysis and gradient optimization software.

## 2. Results and Discussion

Phosphodiester stationary phases represent a group of polar-embedded materials. Structures are presented in Figure 1. The presence of a phosphate group and hydrophobic ligand allows for the retention of compounds in RP LC and HILIC. Thus, β-blockers are interesting compounds due to their various polarities and variety of functional groups. The presence of hydrophobic and polar groups allows the investigation of the selectivity of phosphodiester stationary phases in different liquid chromatography modes. The phosphate group in the ligand structure has pKa around 1.45 ± 0.5. Thus, phosphate groups are ionized in the mobile phase pH equal to 7.5 and constitute cation exchange sites. The ionized groups are presented in Figure 1. Thus, despite polar and hydrophobic properties, phosphodiester stationary phases are also weak cation exchangers.

Tested β-blockers exhibit pKa values in the range of 9.2–9.7 (details are listed in Table 1). The pH of the mobile phase was 7.5. It means that the hydrogen cation concentration was two orders of magnitude higher. Thus, in the analysis conditions, all compounds are protonated (see Figure 2). As a result, despite hydrophobic interactions in RP LC and hydrophilic interactions in HILIC, β-blockers can ion-exchange with the stationary phase surface. Anytime, in RP LC and HILIC, we observe a mixed-mode retention mechanism. Various types of interactions complicate the optimization of separation conditions but offer different separation possibilities.

### 2.1. Retention Investigation

All β-blockers exhibit a U-shape of retention behavior over the mobile phase composition. The plots of retention factor (k) are presented in Figure 3. First, on all columns, all β-blockers exhibit significant retention, independent of the mobile phase composition. Usually, for polar-embedded stationary phases, for mobile phase composition in the range of 45–55% of organic modifier, there is a minimum of retention. In many cases, compounds in such conditions are eluted in or near the column void volume. Here, we can observe a retention factor of around five (for Diol-P-C-Benzyl and Diol-P-C18). It increases to around k = 7 for Diol-P-Chol (for a wide range of 30–70% of ACN). The lowest retention (around k = 4) is observed for Diol-P-C10. Detailed data on retention factors are listed in Appendix A.

In the ACN concentration range of 30–70% in ammonium acetate, the retention of all β-blockers is similar, which makes separation impossible (Figure 3). It is a result of the cation-exchange mechanism. Protonated molecules interact with ionized phosphate groups that result in retention but do not offer significant selectivity.

The decreasing or increasing ACN concentration out of this range enormously increases retention. The retention varies between RP LC and HILIC depending on the stationary phase. Diol-P-Benzyl and Diol-C10 stationary phases provide higher retention in HILIC compared to the RP LC. In contrast, the retention on Diol-P-Chol and Diol-P-C18 exhibits comparable retention properties in both RP LC and HILIC (Figure 3). Generally, the highest retention factor was observed for Diol-P-Chol on both sides, RP and HILIC; however, atenolol in HILIC has the highest retention on Diol-P-Benzyl. It results from higher carbon load and surface coverage density of Diol-P-Chol stationary phases (see Table 2). Lower surface coverage and resulting higher accessibility to the silanol group on Diol-P-Benzyl and Diol-PC10 probably cause the domination of the HILIC mechanism.

Specific retention of atenolol on Diol-P-Benzyl in HILC is a result of two factors. First, the Diol-P-Benzyl stationary phase has the lowest coverage of hydrophobic groups and higher accessibility to hydroxyl groups that increase the HILIC retention mechanism. Second, atenolol has the lowest hydrophobicity (see Table 1) and the highest hydrophilicity, which allows strong retention in HILIC.

Changes in the particular β-blockers’ retention depend on their structure. Usually, more polar compounds exhibit higher retention in the HILIC range than in RP LC. On the other hand, more hydrophobic molecules exhibit higher retention in RP LC and lower in HILC. Thus, the retention order is usually the opposite between RP LC and HILIC. In the case of β-blockers, molecules possess both hydrophobic and polar groups (see Figure 2). It causes some of them to have similar retention in RP LC and HILIC, for example, acebutolol. On the other hand, atenolol significantly changes the retention order between HILIC and RP LC.

In RP LC, retention is governed mainly by hydrophobic interactions between the stationary phase and the solute. As evidence, the dependence of log k of particular compounds plotted against its log P value is linear. In the case of the polar-embedded stationary phases, the surface is heterogenous and possesses hydrophobic and polar adsorption sites. If the solute is polar, the retention mechanism is governed mainly by polar interactions (e.g., hydrogen bonds). The number of hydrogen bond donors and acceptors for each β-blocker is listed in Table 1. However, hydrophobic interactions may also occur but to a lower extent. These polar (hydrophilic) interactions are responsible for retention in HILIC. However, they cannot be omitted in reversed-phase conditions. As a result, the dependence of log k vs. log P is nonlinear for β-blockers on tested stationary phases at 10% ACN in the mobile phase. Detailed results are presented in Figure 4. Nevertheless, the linear dependence log k vs. log P slope confirms the hydrophobicity of the stationary phases. Higher hydrophobicity of the stationary phases provides a greater value of the curve slope. The most hydrophobic, in order, are Diol-P-Chol and Diol-P-C18, and the weakest hydrophobicity is exhibited by Diol-P-Benzyl.

Deviations from the trend line in Figure 4 result from specific interactions between the solute and the stationary phases. For example, acebutolol (compound F in Figure 4) exhibits retention significantly higher than other compounds resulting from its hydrophobicity. Comparing the chemical structure, acebutolol possesses the highest ability for hydrogen bond creation with the stationary phase (see Table 1), significantly impacting its retention (Figure 2). On the other hand, mexiletine (compound D in Figure 4) provides lower retention than predicted from the trend, which may result from the weak ability for polar interaction compared with other tested β-blockers. Mexiletine has the lowest hydrogen bond donor and acceptor groups (see Table 1)

The opposite situation is observed in Figure 5. The dependence of retention (Log k) on hydrophobicity is declining. This is very logical because, in HILIC, the retention increases with the hydrophilicity of the molecule and decreases when the hydrophobicity increases.

Nevertheless, it should be noted that despite hydrophilic interaction in HILIC and hydrophobic interactions in RP LC, β-blockers are retained mainly by the cation-exchange mechanism.

The presence of cation-exchange properties was confirmed through attempts to elute β-blockers in the ACN-water mobile phase without salt addition at an apparent pH of around 6.8. This pH does not change the form of molecules’ protonation nor the ionization of the stationary phase. However, any of the tested compounds were eluted from the stationary phase. Salt addition provides counterions that allow the protonated β-blockers’ elution from the phosphodiester stationary bonded phases according to the cation-exchange mechanism. It confirms that tested mixed-mode stationary phases exhibit weak cation-exchange properties. The detailed investigation of the cation-exchange mechanism was not the topic of this study. It will be continued in future work.

### 2.2. Separation

The separation of very similar compounds may be a difficult task. Chromatographic resolution depends on three factors: retention, selectivity parameter, and column (system) efficiency, which is measured as a number of theoretical plates. Two of them are crucial, the separation factor and efficiency. High column efficiency in modern UHPLC systems enormously improves chromatographic resolution. However, the selectivity offered by the stationary phase (or mobile phase) is critical.

The present study tested four homemade columns according to chromatographic selectivity in RP LC and HILIC modes. Unfortunately, the efficiency of tested columns was up to 70,000 theoretical plates per meter. Cation-exchange properties cause peak broadening. It reduces efficiency, which results in the loss of resolution. However, the obtained result compares the selectivity of various functionalities bonded as a stationary phase. The selectivity of the separation in RP LC (10% of ACN in ammonium acetate) and for HILIC (90% of ACN in 10 mM ammonium acetate) are presented in Table 3.

Comparing the selectivity in RP (10% ACN) and HILIC (90% ACN), it is easy to conclude that overall selectivity is higher in HILIC. It is observed for all stationary phases, even for more hydrophobic ones such as Diol-P-Chol.

The surprise is that for Diol-C-10 and Diol-P-Chol, atenolol and acebutolol provide the highest retention in both 10% and 90% of ACN. These compounds behave with relatively low hydrophobicity, so the highest retention in RP LC must be governed not only by hydrophobic interactions typical for RP but also due to some polar interactions with the stationary phase surface and cation-exchange mechanism. It confirms the mixed-mode retention model on the tested stationary phase.

### 2.3. Intelligent Peak Deconvolution Analysis

Insufficient separation of target compounds resulting from insufficient selectivity or low efficiency is a significant problem for chromatography. However, if it is possible to collect 3D data, for example, from a PDA detector, data analysis allows separating signals of unseparated peaks. One of them is Intelligent Peak Deconvolution Analysis. Details on the algorithm were described in [24]. According to the literature [24], the algorithm provided less than ±1.0% error between true and separated peak area values.

During the study, i-PDeA II was applied to keep track of unseparated peaks during gradient separation. The exemplary results are listed in Figure 6. In such conditions, oxprenolol and pindolol elute together and provide one symmetrical peak. After performing the deconvolution, two peaks may be determined. According to Table 1, the selectivity of these two compounds equals only 1.01.

The most significant advantage of the deconvolution function is that peaks that are not physically separated can be digitally separated. It reduces the time needed for the analysis of peak tracking in method development and column characterization procedures.

Deconvolution is not limited to two unseparated peaks. In Figure 7, a part of the chromatogram is presented where the blue line presents a measured signal that contains two bands (A and C). Part C is a zoom of part A. After deconvolution, in two signals, five chromatographic peaks were found. Their corresponding spectra are shown in Figure 7B,D. The effect of deconvolution is reduced if the unseparated substances do not differ in the spectra. As seen in Figure 7, if the spectra are different, several peaks may be digitally separated.

### 2.4. Gradient Optimization

For all columns, gradient analyses were optimized in RP LC and HILIC. The best separation was obtained on the Diol-P-C18 stationary phase in RP LC conditions. The resulting chromatogram is presented in Figure 8. Only two compounds, mexiletine and pindolol, were not fully separated. However, it was overcome by deconvolution analysis. The result shows that β-blockers may be separated on homemade columns using a mixed-mode retention mechanism. The further increase in column efficiency may significantly improve the resolution. It confirms that phosphodiester stationary phases are unique and promising chromatographic materials.

## 3. Materials and Methods

### 3.1. Materials and Reagents

Four house-made stationary phases were tested during the study. These materials contain different hydrophobic groups bonded to diol-silica by a phosphate group. The structures of chemically bonded phases are presented in Figure 1.

Detailed characteristics of these materials are presented in the previous studies [25,26]. Kromasil 100 silica gel (Akzo Nobel, Bohus, Sweden) was used as a support for the stationary phase synthesis. The properties of the stationary phases are listed in Table 2. Stationary phases were packed into 125 × 4.6 mm i.d. stainless steel columns using laboratory-made equipment and a Haskel (Burbank, CA, USA) packing pump using the slurry method.

Acetonitrile was high-purity “for HPLC” gradient grade, and ammonium acetate was “for HPLC” from Sigma-Aldrich (St. Louis, MO, USA). Water was purified using a Milli-Q system (Millipore, El Paso, TX, USA) in our laboratory. Mobile phases consisted of acetonitrile and 10 mM ammonium acetate in water, with pH equal to 7.5 (adjusted with 1M ammonium hydroxide solution).

### 3.2. Compounds

Eight beta-adrenergic blockers were used in the study. The chemical structures are presented in Figure 2. The hydrophobicity of the compounds, measured as log P value and pK_a_ values, are listed in Table 2. Compounds were dissolved in HPLC water. The concentration of the samples was 1 mg/mL.

### 3.3. Instruments

All the experiments were conducted on the Shimadzu Prominence system (Kioto, Japan). This instrument includes a quaternary solvent delivery pump (LC-20AD) with an online degasser, an autosampler (SIL-20A), a column thermostat (CTO-10 AS VP), a spectrophotometric diode-array UV-Vis detector (SPD-M20A), and a data acquisition station. The data were collected in LabSolutions software.

Peak Deconvolution Analysis (Intelligent Peak Deconvolution Analysis i-PDeA II) was applied to track unseparated peaks. i-PDeA II is a part of LabSolutions software by Shimadzu Corporation (Kioto, Japan) [21,22,24].

ChromSword (Riga, Latvia) software was used for optimizing the gradient conditions [27,28].

### 3.4. Methods

All the measurements were undertaken with the mobile phase’s 1 mL/min flow rate. The column thermostat was set to 30 °C, while the autosampler temperature was set to 5 °C. The injection volume was 1 µL for the analysis of single components and 10 µL for mixtures. Measurements were made in triplicate.

The gradient profiles were modeled in an off-line mode basis on the retention data. As an input, the parameters (retention time, peak area, and peak width at 50% high) of two linear gradients from 0 to 50% of a given solvent (acetonitrile in RP LC and 10 mM ammonium acetate in HILIC) in different times were used.

## 4. Conclusions

Using phosphodiester stationary phases, β-blockers exhibit retention in the RP LC and HILIC range of mobile phase composition. Their retention plot over the mobile phase composition demonstrates a characteristic U shape. The retention mechanism in RP LC is based mainly on the compound hydrophobicity; however, polar interactions play a significant role. It was also confirmed that phosphodiester stationary phases exhibit weak cation-exchange properties. Eluting β-blockers without salt added to the mobile phase is impossible. Unfortunately, cation-exchange properties cause band broadening and reduce the separation resolution, which may be overcome by applying gradient elution.

Applying Intelligent Peak Deconvolution Analysis may solve the problem with the co-elution of particular compounds. Deconvolution allows the digital separation of peaks based on their spectra using a PDA detector. Intelligent Peak Deconvolution Analysis is a promising tool facilitating the optimization of chromatographic methods.

Analyses of β-blockers allow describing the mixed-mode and cation-exchange properties of phosphodiester stationary phases. The best separation was obtained on Diol-P-C18 in gradient elution. Obtained results show the versatility of tested stationary phases for their application in RP LC, HILIC, and as a weak cation exchanger.

This article presents a series of preliminary studies on applying novel materials in RP LC and HILIC. Further investigation will focus on the cation-exchange mechanism, description, and optimization.

## Figures and Tables

**Figure 1 molecules-28-03249-f001:**
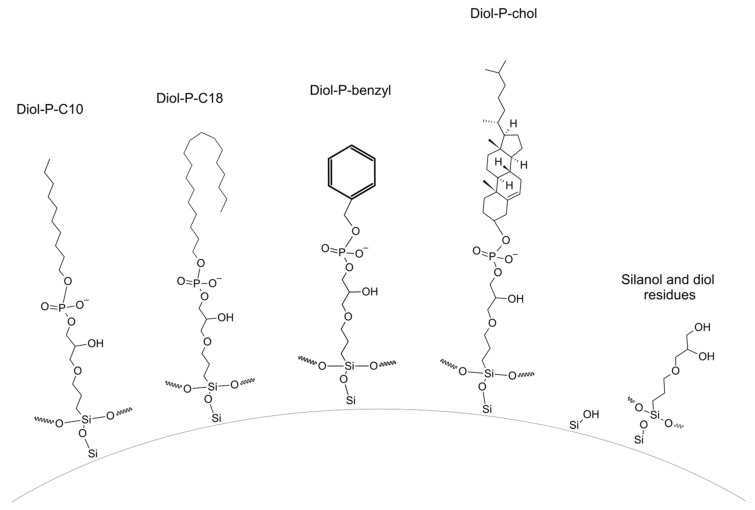
Structures of chemically bonded stationary phases used in the study at pH = 7.5.

**Figure 2 molecules-28-03249-f002:**
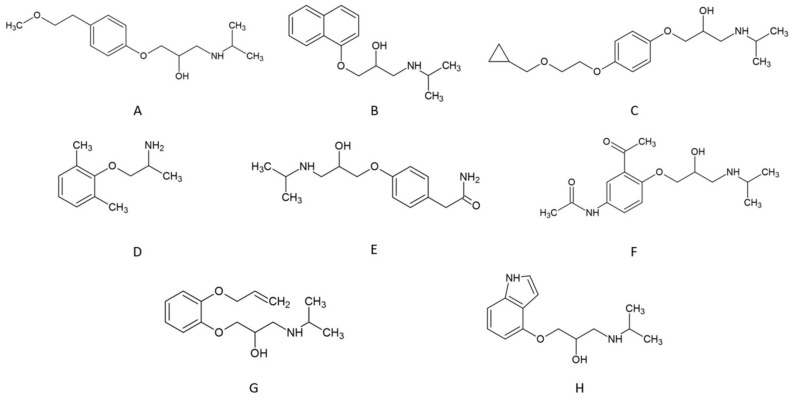
Structure of tested β-blockers at pH equal 7.5: (**A**)–metoprolol, (**B**)–propranolol, (**C**)–ciciloprolol, (**D**)–mexiletine, (**E**)–atenolol, (**F**)–acebutolol, (**G**)–oxprenolol, and (**H**)–pindolol.

**Figure 3 molecules-28-03249-f003:**
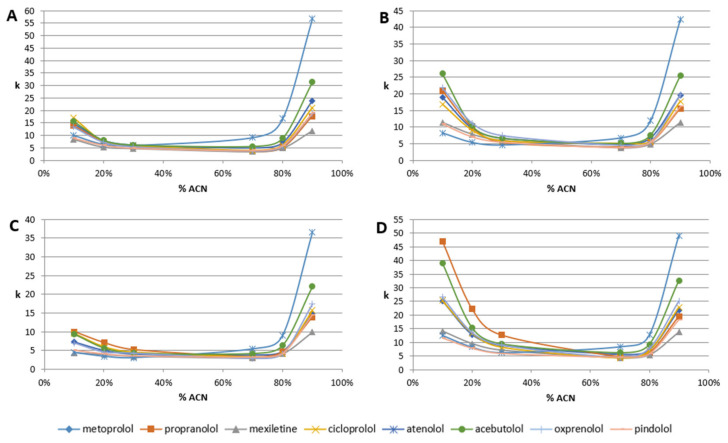
Retention factor (k) dependence for different mobile phase composition (%ACN in 10 mM ammonium acetate in water, pH equal 7.5); (**A**)-Diol-P-Benzyl, (**B**)-Diol-P-C18, (**C**)-Diol-P-C10, and (**D**)-Diol-P-Chol.

**Figure 4 molecules-28-03249-f004:**
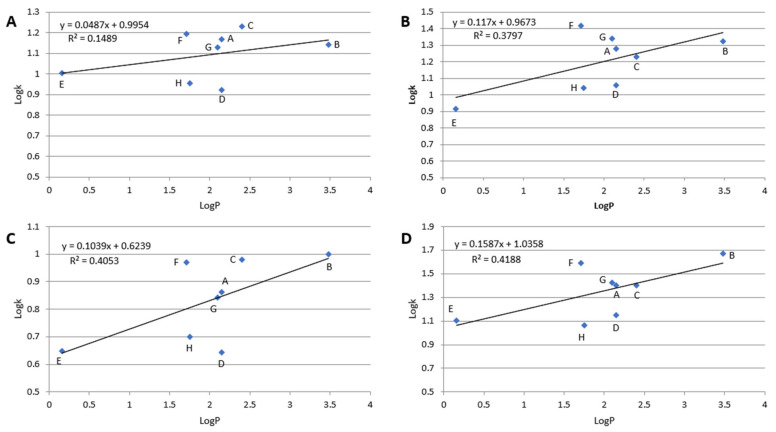
Dependences of log k vs. log P; (**A**)–Diol-P-Benzyl, (**B**)–Diol-P-C18, (**C**)–Diol-P-C10, and (**D**)–Diol-P-Chol measured for 10% ACN in 10 mM ammonium acetate in water. (A)–metoprolol, (B)–propranolol, (C)–ciciloprolol, (D)–mexiletine, (E)–atenolol, (F)–acebutolol, (G)–oxprenolol, and (H)–pindolol.

**Figure 5 molecules-28-03249-f005:**
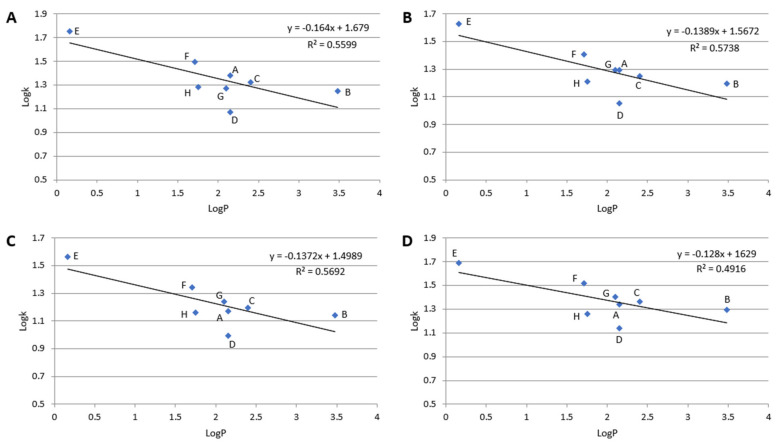
Dependences of log k vs. log P; (**A**)–Diol–P–Benzyl, (**B**)–Diol–P–C18, (**C**)–Diol–P–C10, and (**D**)–Diol–P–Chol measured for 90% ACN in 10 mM ammonium acetate in water. (A)–metoprolol, (B)–propranolol, (C)–ciciloprolol, (D)–mexiletine, (E)–atenolol, (F)–acebutolol, (G)–oxprenolol, and (H)–pindolol.

**Figure 6 molecules-28-03249-f006:**
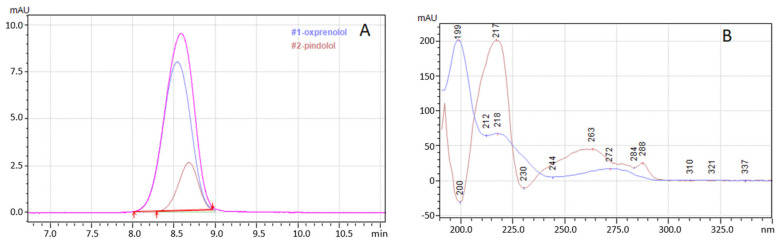
Deconvolution of unseparated oxprenolol and pindolol on Diol-P-Benzyl column in 90% of ACN (**A**) and corresponding UV spectra (**B**). Pink line represent signal obtained from detector.

**Figure 7 molecules-28-03249-f007:**
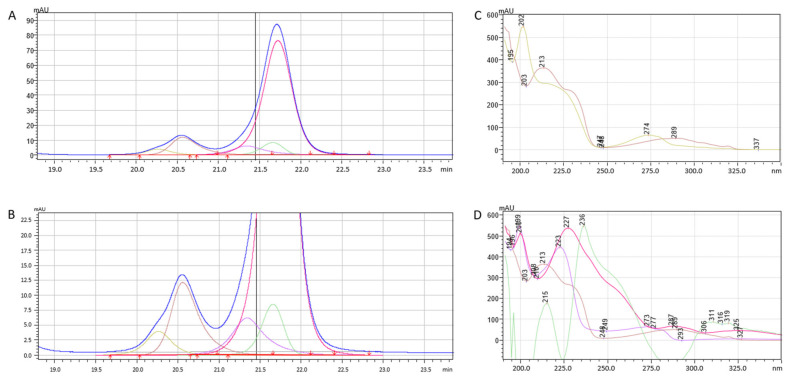
Deconvolution of unseparated compounds (**A**,**B**) and corresponding UV spectra (**C**,**D**). Deconvoluted signal of various compounds are plotted in different colors whereas navy-blue line represents the signal from detector.

**Figure 8 molecules-28-03249-f008:**
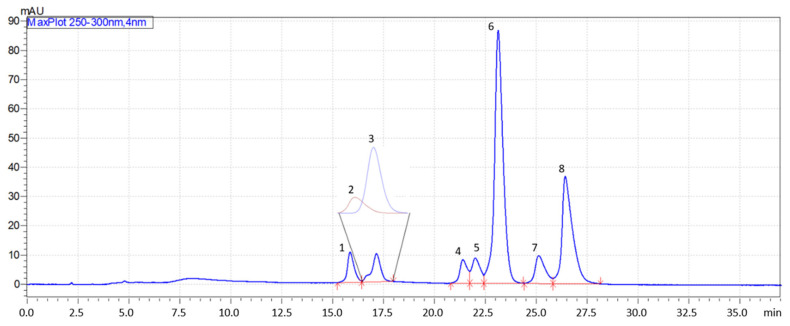
Exemplary chromatogram of optimized gradient separation on Diol–P–C18 stationary phase; linear gradient from 0% to 50% of ACN in 10 mM ammonium acetate; 1–atenolol, 2–mexiletine, 3–pindolol, 4–metoprolol, 5–oxprenolol, 6–acebutolol, 7–cicloprolol, 8–propranolol; compounds 2 and 3 were separated using peak deconvolution.

**Table 1 molecules-28-03249-t001:** Characteristics of compounds used in the study.

Abbreviation	Beta-Blocker	Number of Hydrogen Bonds Donor	Number of Hydrogen Bonds Acceptor	Log P *	pK_a_ *
A	metoprolol	2	4	2.15	9.56–9.70
B	propranolol	2	3	3.48	9.53–9.45
C	cicloprolol	2	5	2.40	9.2
D	mexiletine	1	2	2.15	9.14–9.15
E	atenolol	3	5	0.16	9.54–9.60
F	acebutolol	3	6	1.71	9.52–9.67
G	oxprenolol	2	4	2.10	9.57
H	pindolol	3	4	1.75	9.25–9.54

* Data according to [23].

**Table 2 molecules-28-03249-t002:** Characteristics of stationary phases used in the study.

Stationary Phase	Carbon Load [%]	Coverage Density [µmol/m^2^]
Diol-P-C10	3.43	0.56
Diol-P-C18	4.18	0.42
Diol-P-Benzyl	2.86	0.56
Diol-P-Chol	9.31	0.87

**Table 3 molecules-28-03249-t003:** Selectivity in RP (10% ACN) and HILIC (90% ACN).

Column	Pair	10% ACN	Pair	90% ACN
Diol-P-C10	mexiletine/atenolol	1.01	mexiletine/propranolol	1.41
	atenolol/pindolol	1.13	propranolol/pindolol	1.04
	pindolol/metoprolol	1.39	pindolol/metoprolol	1.02
	metoprolol/cicloprolol	1.05	metoprolol/cicloprolol	1.06
	cicloprolol/oxprenolol	1.28	cicloprolol/oxprenolol	1.11
	oxprenolol/acebutolol	1.02	oxprenolol/acebutolol	1.27
	acebutolol/atenolol	1.04	acebutolol/atenolol	1.65
Diol-P-C18	atenolol/pindolol	1.34	mexiletine/propranolol	1.38
	pindolol/mexiletine	1.04	propranolol/pindolol	1.03
	mexiletine/cicloprolol	1.48	pindolol/cicloprolol	1.10
	cicloprolol/metoprolol	1.12	cicloprolol/metoprolol	1.11
	metoprolol/propranolol	1.10	metoprolol/oxprenolol	1.00
	propranolol/oxprenolol	1.05	oxprenolol/acebutolol	1.29
	oxprenolol/acebutolol	1.19	acebutolol/atenolol	1.66
Diol-P-Benzyl	mexiletine/pindolol	1.07	mexiletine/propranolol	1.51
	pindolol/atenolol	1.12	propranolol/oxprenolol	1.06
	atenolol/oxprenolol	1.33	oxprenolol/pindolol	1.01
	oxprenolol/propranolol	1.04	pindolol/cicloprolol	1.11
	propranolol/metoprolol	1.06	cicloprolol/metoprolol	1.13
	metoprolol/acebutolol	1.06	metoprolol/acebutolol	1.31
	acebutolol/cicloprolol	1.09	acebutolol/atenolol	1.81
Diol-P-Chol	pindolol/atenolol	1.09	mexiletine/pindolol	1.33
	atenolol/mexiletine	1.12	pindolol/propranolol	1.07
	mexiletine/metoprolol	1.78	propranolol/metoprolol	1.12
	metoprolol/cicloprolol	1.00	metoprolol/cicloprolol	1.05
	cicloprolol/oxprenolol	1.05	cicloprolol/oxprenolol	1.10
	oxprenolol/acebutolol	1.47	oxprenolol/acebutolol	1.29
	acebutolol/atenolol	1.20	acebutolol/atenolol	1.50

## Data Availability

The data presented in this study are available on request from the corresponding author.

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
