# Peer review of "Beta-Blocker Separation on Phosphodiester Stationary Phases—The Application of Intelligent Peak Deconvolution Analysis"

_molecules, 2023, doi:10.3390/molecules28073249_

Round 1

Reviewer 1 Report

Beta-blockers are a class of drugs that are used to treat abnormal heart rhythms. Meanwhile, long-term side effects have been reported as well. Due to the importance of beta-blockers, the authors used different stationary phases to separate beta-blockers more efficiently and selectively. However, the study was not well explained due to several reasons:

1.    In Figure 3, the authors claimed that Diol-P-Chol (k=10) had the highest retention factor among all four stationary phases and Diol-P-Benzyl gave the lowest retention factor. However, based on the curves, they all looked very similarly, especially between 40-60% of ACN. Maybe use a zoomed-in figure or table to better explain that.

2.     In Figure 3, the authors explained that atenolol had the highest retention factor in HILIC on Diol-P-Benzyl because of carbon load and coverage density. If that’s the case, would other beta-blockers also have higher retention factors on Diol-P-Benzyl? Is there any way to increase the carbon load for Diol-P-Benzyl?

3.     In Figure 4 and 5, please include R2 values in the plots.

4.     The molar mass column in Table 1 was never mentioned in the paper. Not sure why it was there. The pKa values were only used once.

5.     The authors also mentioned that hydrogen bonding interactions could impact the retention of beta-blockers. It would be helpful to have a column in Table 1 to list the amount of hydrogen bond donor and acceptor of each beta-blocker.

6.     Figure 8 gave a nice separation of all 8 compounds. Why did 6 and 8 have much higher peaks than the others?  

7.     The scientific writing needs to be improved. For example, in Table 1, logP value should be 2.15 instead of 2,15.

8.     The paper can benefit from language editing for better clarity.

Author Response

  1.  In Figure 3, the authors claimed that Diol-P-Chol (k=10) had the highest retention factor among all four stationary phases and Diol-P-Benzyl gave the lowest retention factor. However, based on the curves, they all looked very similarly, especially between 40-60% of ACN. Maybe use a zoomed-in figure or table to better explain that.
    Response: Yes, in the range 40-60% ACN only Diol-P-Chol column provide visible higher retention. We analyzed data and made same correction in the text. We add Table S1 that contain all retentin data. 
  2. In Figure 3, the authors explained that atenolol had the highest retention factor in HILIC on Diol-P-Benzyl because of carbon load and coverage density. If that’s the case, would other beta-blockers also have higher retention factors on Diol-P-Benzyl? Is there any way to increase the carbon load for Diol-P-Benzyl?
    Response: No, other beta-blockers have retention comparable with Diol-P-Chol. 
    Probably it is possible to get higher surface coverage; however, it is the best that we have obtained. On the other hand, low surface coverage with hydrophobic groups provides good HILIC properties. It may be an advantage. 
  3. In Figure 4 and 5, please include Rvalues in the plots.
    Response: Figures are modified. R2 is provided. 
  4. The molar mass column in Table 1 was never mentioned in the paper. Not sure why it was there. The pKa values were only used once.
    Response: Molar mass was removed from Table 1.
  5. The authors also mentioned that hydrogen bonding interactions could impact the retention of beta-blockers. It would be helpful to have a column in Table 1 to list the amount of hydrogen bond donor and acceptor of each beta-blocker.
    Response: Data were added to Table 1.
  6. Figure 8 gave a nice separation of all 8 compounds. Why did 6 and 8 have much higher peaks than the others?  
    Response: All compounds have the same concentration. The difference is a result of different UV light absorption. 
  7. The scientific writing needs to be improved. For example, in Table 1, logP value should be 2.15 instead of 2,15.
    Response: It was corrected. We are sorry for the mistake. We are using commas instead of points.
  8. The paper can benefit from language editing for better clarity.
    Response: English was corrected in the whole manuscript.

Reviewer 2 Report

This research article is aiming to apply several columns using phosphodiester stationary phases coupled with Intelligent Peak Deconvolution Analysis to separate beta-blockers. Although the method design is straightforward, this work needs a major revision on its language style to present their findings better. Please see specific suggestions below.

Comments:

1. P.1 line 12, (RPLC).

2. P.2 line 57, not a strong connection between sentences. “…while some of them are completely protonated…”, add “while”?

3. P.2 line 61, “reduced to 5 minutes”, are these compounds fully separated? If so, please add this sentence.

4. P.2 line 69, “Depending on…”, this is not a correct sentence in its grammar. 

5. P.2 line 70, “…it may be ionized”, please indicate the subject. “it” is “such materials”? If so, should be “they”.

6. P.2 line 72, please rephrase “in water as a mobile phase”.

7. P.2 line 73, please specify the subject instead of using “such materials”.

8. P.2 line 75, “natures”? What’s the difference between “natures” and structures?

9. P.2 line 85, please rephrase and double check the subject/object when using “constitute”.

10. P.2 line 88, “are interesting compounds due to their structure” sounds cryptic, please rephrase. 

Author Response

We would like to thank the reviewer very much for pointing out the errors enabling us to increase the quality of our work.

English was corrected in the whole manuscript. 

Round 2

Reviewer 1 Report

Just a quick follow-up question based on Q2.  Do you have any potential explanations on why atenolol had the highest retention factor, but other beta-blockers had comparable retention factors with Diol-P-Chol? 

Scientific writing can be improved. For example, in Figure 4 and 5, it should be 0.0487 instead of 0,0487.

Author Response

Just a quick follow-up question based on Q2.  Do you have any potential explanations on why atenolol had the highest retention factor, but other beta-blockers had comparable retention factors with Diol-P-Chol? 

Response: 

Specific retention of atenolol on Diol-P-Benzyl in HILC is a result of two factors. First, Diol-P-Benzyl stationary phase has lowest coverage of hydrophobic groups and higher accessiblity to hydroxyl groups that increase HILIC retention mechanism. Second, atenolol has the lowes hydrophobictiy and the highest hydrophilicity that allow strong retention in HILIC.

It is added to the manuscript.

Scientific writing can be improved. For example, in Figure 4 and 5, it should be 0.0487 instead of 0,0487.

Response: Figures are corrected. We are sorry for the mistake; we use commas for everyday work.

Reviewer 2 Report

Thank you for addressing these questions and it looks better. 

Author Response

We thank the Reviewer for ideas to improve our work.